# Absolute concentration estimation of COVID-19 convalescent and post-vaccination IgG antibodies

**Melvin E. Klegerman**[1]*, **Tao Peng**[1], **Ira Seferovich**[2], **Mohammad H. Rahbar**[3,4,5], **Manouchehr Hessabi**[4], **Amirali Tahanan**[4], **Audrey Wanger**[6], **Carolyn Z. Grimes**[1], **Luis Z. Ostrosky-Zeichner**[1], **Kent Koster**[7], **Jeffrey D. Cirillo**[7], **Dinuka Abeydeera**[2], **Steve De Lira**[2], **David D. McPherson**[1]

1 Department of Internal Medicine, McGovern Medical School, The University of Texas Health Science Center at Houston, Houston, TX, United States of America, 2 Carterra, Inc., Salt Lake City, UT, United States of America, 3 Department of Epidemiology, Human Genetics, and Environmental Sciences (EHGES), School of Public Health, The University of Texas Health Science Center at Houston, Houston, TX, United States of America, 4 Biostatistics/Epidemiology/Research Design (BERD) Component, Center for Clinical and Translational Sciences (CCTS), The University of Texas Health Science Center at Houston, Houston, Texas, United States of America, 5 Division of Clinical and Translational Sciences, Department of Internal Medicine, McGovern Medical School, The University of Texas Health Science Center at Houston, Houston, TX, United States of America, 6 Department of Pathology and Laboratory Medicine, McGovern Medical School, The University of Texas Health Science Center at Houston, Houston, TX, United States of America, 7 Department of Microbial Pathogenesis and Immunology, Texas A&M University Health Science Center, Bryan, TX, United States of America

* Melvin.E.Klegerman@uth.tmc.edu

**Data Availability Statement:** ELISA affinity calculations, plasma sample results, and sample collection data are available from the Dryad

## Abstract

Soon after commencement of the SARS-CoV-2 disease outbreak of 2019 (COVID-19), it became evident that the receptor-binding domain of the viral spike protein is the target of neutralizing antibodies that comprise a critical element of protective immunity to the virus. This study addresses the relative lack of information regarding actual antibody concentrations and binding affinities in convalescent plasma (CP) samples from COVID-19 patients and extends these analyses to post-vaccination (PV) samples to estimate protective IgG antibody (Ab) levels. A direct enzyme-linked immunosorbent assay (ELISA) was used to measure IgG anti-spike protein (SP) antibodies (Abs) relative to human chimeric spike S1 Ab standards. Microplate wells were coated with recombinant SP. Affinities of Ab binding to SP were determined by previously described methods. Binding affinities were also determined in an RBD-specific sandwich ELISA. Two indices of protective immunity were determined as permutations of Ab molar concentration divided by affinity as dissociation constant ($K_D$). The range and geometric means of Ab concentrations in 21 CP and 21 PV samples were similar and a protective Ab level of 7.5 µg/ml was determined for the latter population, based on 95% of the normal distribution of the PV population. A population (n = 21) of plasma samples from individuals receiving only one vaccination with the BNT162b2 or mRNA-1273 vaccines (PtV) exhibited a geometric mean Ab concentration significantly (p < 0.03) lower than the PV population. The results of this study have implications for future vaccine development, projection of protective efficacy duration, and understanding of the immune response to SARS-CoV-2 infection.

repository through the URL https://doi.org/10.5061/dryad.n5tb2rc4t.

**Funding:** This study was funded by the following sources: Texas A&M University System, BADAS Study (Bacillus Calmette-Guérin Vaccination as defense against SARS-CoV-2. A Randomized Placebo-Controlled Trial to Protect Health Care Workers by Enhanced Trained Immune Responses) (MEK, JDC, KK). Subcontract; JDC is the PI of the parent grant. Biostatistics/Epidemiology/Research Design (BERD) component of the Center for Clinical and Translational Sciences (CCTS), mainly funded by a grant (UL1 TR003167) from the National Center for Advancing Translational Sciences (NCATS), awarded to The University of Texas Health Science Center at Houston (UTHealth) (MHR, MH, AT). DDM is the grant PI. University of Texas Health Science Center, McGovern Medical School Department of Internal Medicine Pilot Grant (Post-Vaccination Antibody Characterization) (MEK, DDM). DDM is the Chair of the Department. The funders had no role in study design, data collection and analysis, decision to publish, or preparation of the manuscript.

**Competing interests:** The authors have declared that no competing interests exist.

## Introduction

The coronavirus disease outbreak of 2019 (COVID-19), caused by a newly emerging coronavirus, severe acute respiratory syndrome coronavirus 2 (SARS-CoV-2), has infected more than 773 million people worldwide, resulting in nearly 7 million deaths (more than 1.1 million in the U.S.) four years after initial cases appeared in Wuhan, China, in December 2019 [1]. The outstanding structural feature of coronaviruses (from which they derive their crown-like designation) is the protruding spikes that mediate attachment of the spherical virions to host cells and subsequent fusion with epithelial cell membranes, required for entry and infection. The spike glycoprotein (SP) that forms these structures is a homotrimer consisting of two subunits, S1 and S2. The S1 protein binds to the angiotensin converting enzyme-2 (ACE-2) receptor on the cell surface through the receptor-binding domain (RBD), while the S2 protein mediates cell membrane fusion [2].

It is readily hypothesized, therefore, that the RBD is the target of neutralizing antibodies that comprise the critical element of protective immunity to the virus. This hypothesis is supported by the finding that the RBD is immunodominant and the target of 90% of the neutralizing activity present in SARS-CoV-2 immune sera vs. the original (wild type) pandemic strain D614G [3]. Most compelling is the established 95% protective efficacy of the two mRNA COVID-19 vaccines developed by Pfizer and Moderna against this strain, which elicit antibodies specific for the RBD [4]. Although many studies have focused on characterization of the antibody response to SARS-CoV-2, emphasis has largely been on antibody properties that are most relevant to effective vaccine development, such as the ability of patients to produce high-affinity IgG antibodies specific for the RBD [2, 5–8], or production of therapeutic monoclonal antibody formulations [3, 9–11]. Studies of actual antibody concentrations and binding affinities in terms of equilibrium constants in convalescent plasma (CP) and post-vaccination (PV) plasma samples have been lacking, with protective antibody levels continuing to be reported as viral neutralizing and antibody binding titers.

We report here the determination of five different parameters addressing these variables in 21 convalescent plasma samples, using quantitative ELISA methodology. We then applied these methods to characterization of 21 plasma samples obtained from individuals who completed the two mRNA vaccination protocols (post-vaccination, PV) and 21 individuals who received only the first vaccination (partial vaccination, PtV). Knowing the protective efficacy of these vaccines, which were virtually identical, we were able to estimate protective IgG anti-SP antibody levels and analogous values for two indexes that combined antibody concentrations and binding affinities for both the SP and the RBD.

## Materials and methods

### Plasma samples

Convalescent plasma (CP) samples were obtained from the University of Texas Health Science Center at Houston (UTHealth)/Memorial Hermann COVID-19 Convalescent Plasma Program under the direction of Dr. Henry E. Wang. Convalescent plasma (CP) donors previously tested positive for COVID-19, were symptom-free for >14 days, tested negative for COVID-19 antigen prior to donation, and tested positive for CP antibodies. Donor plasma was collected by the Gulf Coast Regional Blood Center before August 2020, which created a general stock of CP units that were distributed among therapeutic CP programs in the greater Houston area. Each plasma sample derived from a single donor.

Post-vaccination (PV) plasma samples were obtained from Dr. Luis Ostrosky's laboratory at the Division of Infectious Diseases, Department of Internal Medicine, McGovern School of

Medicine of UTHealth. Two groups of 21 samples, each collected 15–29 days after administration of the first or second dose of the Pfizer or Moderna COVID-19 vaccine between January 4 and February 5, 2021, were studied. Analysis of de-identified convalescent and post-vaccination plasma samples was approved by the UTHealth Committee for Protection of Human Subjects.

## ELISA for human IgG antibodies specific for the SARS-CoV-2 spike protein

This is a direct ELISA in which the antigen is adsorbed directly onto microtiter wells, as we previously described for fibrinogen [12]. Incubation volumes were 50 μl and all incubations except for the initial coating step were at 37˚C. After the blocking step, all incubations were followed by three washes with 0.02 M phosphate-buffered saline, pH 7.4 with 0.05% Tween-20 (PBS-T). Test wells were coated with 5 μg recombinant spike protein (rSP; S1+S2; Creative Diagnostics, Shirley, NY)/ml coating buffer (0.05 M sodium bicarbonate, pH 9.6) overnight at 4˚C. Well contents were aspirated and all wells (including background wells for each sample and standard dilution) were blocked for 1 hour with conjugate buffer (1% bovine serum albumin in 0.05 M Tris, pH 8.0, with 0.02% sodium azide). Human anti-SP IgG standards (chimera, GenScript A02038, Piscataway, NJ) (in serial dilutions of 200–12.5 ng/ml PBS-T) and convalescent plasma samples (3 dilutions within the measurable range, previously determined in a screening assay) were incubated in duplicate for 2 hours. All wells were then incubated with 3,000-fold diluted goat anti-human IgG-alkaline phosphatase conjugate (Sigma-Aldrich, St. Louis, MO) in conjugate buffer for 1 hour. The assay was developed by adding substrate buffer (0.05 M glycine buffer, pH 10.5, with 1.5 mM magnesium chloride) to each well, followed by 4 mg paranitrophenylphosphate (PNPP; Sigma-Aldrich)/ml substrate buffer (for a total volume of 100 μl/well) and incubating for 15 minutes. The reaction was terminated by adding 50 μl 1 M sodium hydroxide to each well. Plates were read at 405 nm wavelength with a BioTek ELx808 multiwell plate reader.

The OD of background wells was subtracted from test well ODs. The net OD of antibody standard wells was plotted vs. IgG antibody concentration, which obeys a hyperbolic relation. For determination of unknown sample antibody concentrations, the curve fit equation was solved for x. Binding affinities with correction for conjugate incubation perturbation were determined as previously described [12]. A correction nomogram for this ELISA is reproduced in S1A Fig. Binding of ACE-2 to the SP was determined by incubating ACE-2 huFc fusion protein (R&D Systems, Minneapolis, MN) during the primary incubation, instead of human antibody. The anti-human AP conjugate binds to the Fc.

## ELISA for human IgG antibodies specific for the SARS-CoV-2 spike protein receptor-binding domain (RBD)

This is a sandwich ELISA in which a capture antibody, rabbit anti-mouse IgG, is adsorbed onto microtititer wells, followed by antigen capture. Test wells were coated with 2,000X diluted rabbit anti-mouse IgG (Sigma-Aldrich) in coating buffer overnight at 4˚C. After blocking, 0.5 μg RSD-mFc (GenScript)/ml PBS-T was added to all wells and incubated for 2 hours. Human anti-SP IgG standards (in serial dilutions of 100–6.25 ng/ml PBS-T) and convalescent plasma samples (3 dilutions within the measurable range, previously determined in a screening assay) were incubated in duplicate for 1 hour. The rest of the ELISA protocol was the same as the rSP ELISA. The ODs of PBS-T only wells run in each assay were subtracted from test well ODs. Calculations of standards and sample antibody concentrations were performed as for the rSP ELISA. A correction nomogram for this ELISA is reproduced in S1B Fig.

## Indices of protective immunity

SP/SP $K_D$ is molar concentration of IgG antibody measured in the spike protein ELISA divided by the antibody $K_D$ measured in that assay. SP/RBD $K_D$ is molar concentration of IgG antibody measured in the spike protein ELISA divided by the antibody $K_D$ measured in the RBD ELISA.

## Carterra LSA RBD ELISA

**Surface preparation.**   An HC200M chip (Carterra PN 4287) was preconditioned via a 2 minute cycling exposure with 50 mM sodium hydroxide (Carterra PN 3638), 500 mM sodium hydroxide, and 10 mM glycine, pH 2.0 (Carterra PN 3640). For goat anti-mouse IgG lawn coupling, the LSA was primed in 25 mM MES, pH 5.5, running buffer + 0.05% Tween-20. The HC200M chip was activated for 8 minutes via cycling exposure of a mixture of 33 mM Sulfo-NHS (Thermo PN 24510), 133 mM EDC (Thermo PN PG82079), 100 mM MES, pH 5.5 (Carterra PN 3625).

Protein coupling was performed by preparing a homogenous anti-murine Fc-specific surface via a 10 minute cycling exposure of 100 μg/ml goat anti-mu Fc specific IgG (Jackson Immunoresearch PN 115-005-071) in 10 mM sodium acetate, pH 4.5 (Carterra PN 3622) + 0.05% Tween-20. The coupling reaction was then quenched by a 6 minute exposure to 1M ethanolamine, pH 8.5 (Carterra PN 3626) + 0.05% Tween-20. Finally, 2 μM RBD-mFc (R&D Systems) was cycled over the anti-mu Fc lawn for 15 minutes.

**Quantitation (titer determination).**   Twenty convalescent plasma samples were prepared between two 96 well plates via serial dilutions of each sample, starting with 10X in 1X HBST, pH 7.4 (Carterra PN 3630), and diluting down to 1,280X. Buffer controls and the human anti-SP IgG standards (6.5–830 nM) were also included in each plate. The LSA's 96 channel printhead sequentially printed the plates on the RBD Lawn. Each plate was cycled via the 96 channel printhead for 30 minutes over the capture surface.

**Blockade (ACE 2 to RBD-Fc).**   After the 30 minute print of each serum plate, the Single Flow Cell (SFC) subsequently injected 588 nM human ACE-2 protein (10-His tag, 85 kDa, R&D Systems) over the chip surface via a 7 minute cycling exposure. The level of ACE-2 to RBD binding was compared between regions that had either serum antibodies bound to RBD or no serum antibodies bound.

## Statistical analysis

As part of descriptive analyses, we examined the distributions of the IgG antibody characteristics (*e.g.*, SP Affinity) for subjects in each of the two groups. If the distribution of an IgG antibody characteristic was skewed, we performed log transformation to produce distributions that better represented a normal distribution for subjects in each of the two groups, convalescent plasma (CP) and post-vaccination (PV). We evaluated the linear associations between each pair of IgG antibody parameters using Pearson correlation coefficients or its nonparametric counterpart measure to examine a linearity. In order to control for probability of type I error, first we used multivariate analysis of variance (MANOVA) to compare all means between IgG antibody characteristics. If a significant difference was found, we used t-tests to identify characteristics for which the means between the two groups were statistically different. We used the Generalized Linear Model (GLM) for assessing potential confounding factors or interactions between the study groups and each of the factors associated with IgG antibody concentrations. Statistical analyses are summarized in Supporting Information (SI file), including S1, S2 and S3 Tables.

## Results and discussion

In the recombinant spike protein (rSP) ELISA, the superimposability of normal human plasma spiked with human anti-SP IgG standard (GenScript, Piscataway, NJ), a measure of assay accuracy, is shown in S2 Fig. Four normal human plasma samples exhibited antibody concentrations of 28.8 ± 13.0 (SE) ng/ml in the assay, which defines the lower limit of specific COVID-19 antibody response, and recoveries of standard from the pooled normal plasma ranged from 94.5% to 132.5% (mean = 109.7%) from 6.25–100 ng Ab/ml. Antibody standard binding affinity for the spike protein, with correction for post-antibody ELISA incubations (12), was found to be 1.46 ± 0.48 nM ($K_D$)(SD, n = 29). Binding affinity of ACE-2 Fc for the spike protein was found to be 2.20 ± 0.60 nM (n = 6), which is similar to published values (2). Dilutions of both convalescent and post-vaccination plasma samples were superimposable on the respective standard curves (Fig 1A and 1B), indicating that the chimera antibody protein standards are suitable for quantitating anti-SP IgG antibodies in human plasma and serum samples.

Anti-SP IgG antibody levels for the convalescent plasma population (n = 21) ranged from 33.1 μg/ml to 1.60 mg/ml, while the corresponding values for the post-vaccination plasma population (n = 21) ranged from 4.9 μg/ml to 1.50 mg/ml (Fig 1C). The geometric means of the two populations were 190 and 176 μg/ml, respectively. The protective antibody plasma concentration, determined as the 5[th] percentile level of the post-vaccination distribution, is 7.5 μg/ml. Anti-SP IgG antibody levels for the partial vaccination population (n = 21) ranged from 0.4 μg/ml to 1.58 mg/ml (Fig 1C). The geometric mean of this population was 70.8 μg/ml, which was significantly lower (p < 0.05) than both the CP and PV populations.

SP binding affinities ($K_D$) of convalescent plasma samples ranged from 0.1 to 2.2 nM (mean = 0.87 ± 0.47 nM), while the range of RBD binding affinities was much narrower, all being subnanomolar (mean = 0.49 ± 0.15 nM). Spike protein binding affinities of post-vaccination plasma IgG antibodies were lower (p = 0.0001) and somewhat broader (0.2–3.9; 1.95 ± 0.99 nM) than those of the convalescent antibody populations. However, the post-vaccination anti-RBD affinities were virtually the same (0.48 ± 0.34 nM; p = 0.877)(Fig 1D). Partial vaccination SP affinities (0.96 ± 0.63 nM) were very similar to the convalescent plasma population, but the RBD affinities were significantly higher ($K_D$ lower) than the convalescent and post-vaccination plasma populations (0.30 ± 0.12 nM; p < 0.01).

The SP/SP $K_D$ index, comprising molar IgG antibody concentration divided by $K_D$, reflects the lower antibody affinities of the post-vaccination population, while the lower mean for the partial vaccination population reflects the lower SP antibody levels (Fig 1E). The similarity of the SP/RBD $K_D$ values for the convalescent and post-vaccination populations reflects the narrow distribution of the RBD affinities. Pertinent data for the convalescent patient and post-vaccination recipient plasma antibody populations are summarized in Table 1.

Based on the criteria of log ranges greater than two and low values in the post-vaccination population below the putative protective level, the most useful indexes appear to be the antibody levels themselves, and the SP/RBD $K_D$ index. This may reflect the narrow range of RBD binding affinities, all of which were higher than that of the ACE-2 protein. Comparison of antibody characteristics between the two populations indicates that natural infection with SARS-CoV-2 confers protective levels of humoral immunity on the survivors. Measures of both the quantity and affinity (binding strength) of IgG antibodies for the whole spike protein were greater in the convalescent patient population, possibly reflecting that the mRNA vaccines produce antibodies more specific for the RBD.

The Carterra LSA RBD ELISA technology yielded titers for 20 convalescent plasma samples ranging from 20.7 to 1,326 (Geom. Mean = 121). A significant linear correlation was found between the LSA RBD titers and the SP IgG antibody levels (r = 0.464, p = 0.039) (S3 Fig), validating the approach by two different methods.

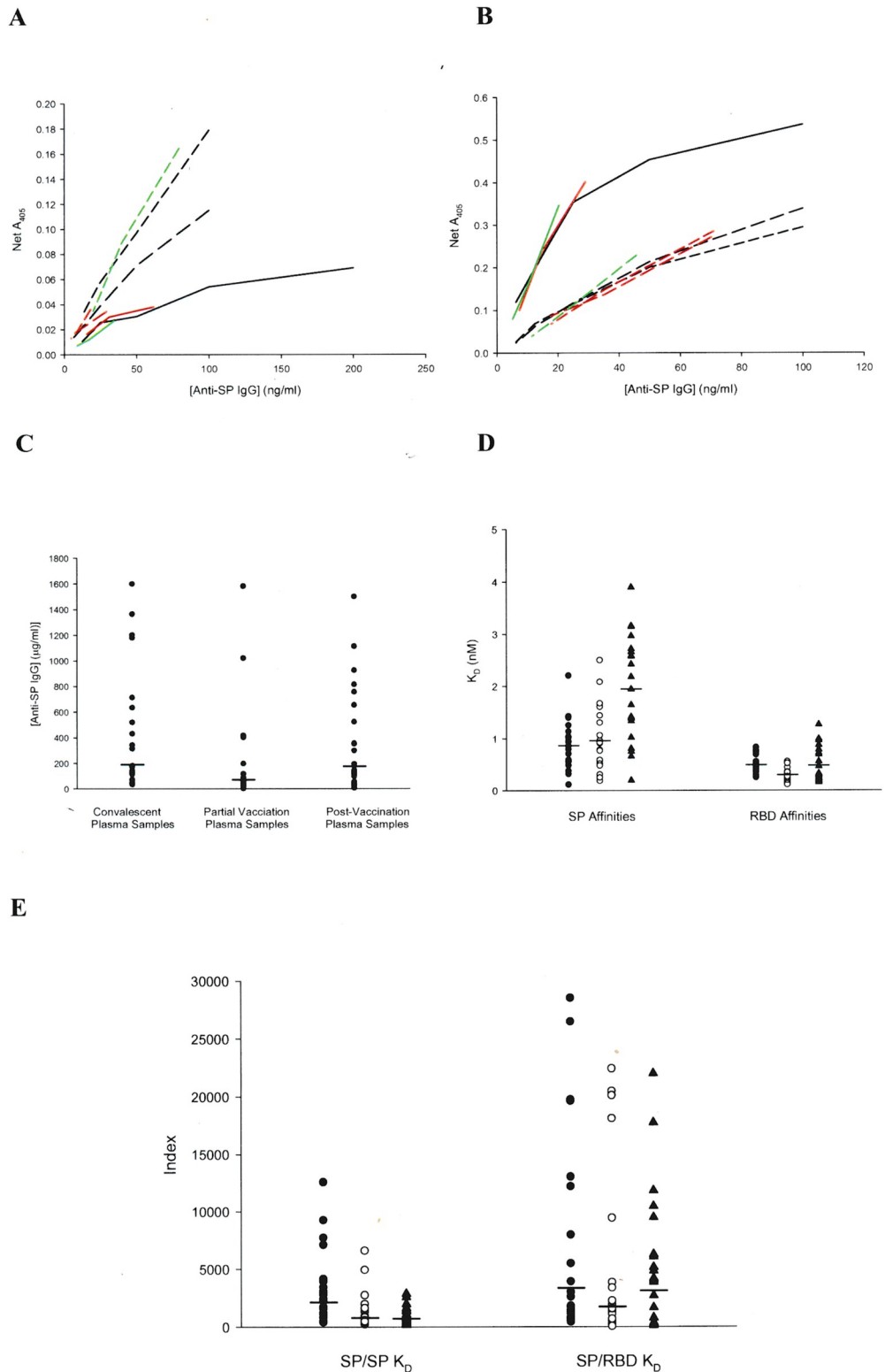

**Fig 1. Superimposability of plasma sample dilution responses on anti-SP IgG standard curves.** Line plots of five samples relative to three corresponding standard curves, defined by averages of two points at each dilution. Colored lines denote samples assayed with the corresponding standards. A. convalescent plasma; B. post-vaccination plasma. Point plots of parameter distributions for convalescent, partial vaccination, and post-vaccination plasma samples. C.

Anti-SP IgG concentrations. Geometric means are indicated by horizontal lines. P < 0.05 for partial vaccination vs. post-vaccination and convalescent plasma samples. D. SP and RBD-specific IgG antibody affinities ($K_D$). Arithmetic means are indicated by horizontal lines. P < 0.01 for SP post-vaccination vs. convalescent and partial vaccination plasma samples, and RBD partial vaccination vs. convalescent and post-vaccination plasma samples. E. Indexes combining plasma IgG antibody concentrations and antibody binding affinities. Geometric means are indicated by horizontal lines. P < 0.005 for SP/SP $K_D$ of convalescent plasma samples vs. partial and post-vaccination plasma samples. Convalescent plasma values are denoted by filled circles, partial vaccination plasma values by open circles, and post-vaccination plasma values by filled triangles in D and E.

Another approach to validation of the anti-SP IgG concentrations is comparison with previously published studies, all of which expressed antibody levels as some form of titer. Relevant data from 11 studies, including the anti-RBD IgG LSA titers reported in this study, are summarized in Table 2. All but one of these studies were characterized by log titer ranges of 1.35 to 4.70. One study of post-vaccination samples, involving immunoassay-determined levels of anti-SP and anti-RBD IgG and neutralization titers against both pseudovirus and live SARS-CoV-2 virus [13], exhibited log ranges for all parameters very close to those found for anti-SP IgG in this study.

Factors for conversion of titers to antibody concentrations (µg/ml), based on equivalence of geometric means are included in Table 2. Neutralization titers show the greatest equivalence, with 4 of 7 ranging from 0.85 to 1.96. The anti-RBD IgG LSA titers tended to be lower than the corresponding anti-SP IgG levels, which is reflected in a regression slope of 0.300 (S3 Fig), and is consistent with the expectation that anti-RBD levels will be lower than anti-SP levels.

In a meta-analysis of neutralization data from seven vaccine studies, Khoury et al. [14] calculated that 50 percent protection against detectable SARS-CoV-1 infection would be afforded by 20.2% of the geometric mean convalescent level of neutralizing antibody. That level for this study is 2.9 µg IgG anti-SP/ml, which is consistent with the 7.5 µg/ml value for 95% protection. Using the conversion factors tabulated in Table 2, the 50% protective values of titers calculated by the Oxford Vaccine Group [13] correspond to a minimum of 9.7µg/ml for anti-RBD IgG.

In order to provide a means of standardizing the many COVID-19 antibody assays that have been utilized, the UK National Institute for Biological Standards and Control (NIBSC) has made available panels of pooled convalescent plasma defined by the WHO International Standard in IU/ml [15]. This standard was utilized in assessing quantitative differences between Moderna (mRNA-1273) vaccine recipients who had COVID-19 breakout infections (n = 36) and those who did not (n = 1,005) [16]. The geometric mean anti-SP IgG level for infected individuals 28 days after the second vaccine dose was 1,890 IU/ml, while the GM for uninfected individuals was 2,652 IU/ml. The corresponding values for anti-RBD IgG were 2,744 and 3,937 IU/ml, respectively. In view of the results of this study, a much larger difference between the two populations would be expected.

**Table 1. Summary of IgG antibody characteristics in convalescent plasma (CP), partial vaccination (PtV), and post-vaccination (PV) samples.**

| Parameter | Log Range | | | Geom. Mean | | | Arith. Mean | | | Prot. Level | Low Value | | |
|---|---|---|---|---|---|---|---|---|---|---|---|---|---|
| | CP | PtV | PV | CP | PtV | PV | CP | PtV | PV | PV | CP | PtV | PV |
| IgG Ab Conc. (µg/ml) | 2.36 | 3.62 | 2.49 | 190 | 70.8 | 176 | — | — | — | 7.5 | 10.6 | 0.4 | 4.9 |
| SP Affinity ($K_D$) (nM) | — | — | — | — | — | — | 0.866 | 0.960 | 1.95 | — | 2.20 | 2.50 | 3.90 |
| RBD Affinity ($K_D$) (nM) | — | — | — | — | — | — | 0.489 | 0.296 | 0.476 | — | 0.820 | 0.550 | 1.26 |
| SP/SP $K_D$ | 1.53 | 1.52 | 1.23 | 2,126 | 791 | 723 | — | — | — | 110 | 374 | 198 | 169 |
| SP/RBD $K_D$ | 1.85 | 2.92 | 2.34 | 3,354 | 1,739 | 3,121 | — | — | — | 161 | 402 | 23.9 | 102 |

**Table 2. Determination of antibody response to SARS-CoV-2 infection and vaccination.**

| Samples | Method/Specificity | | n | Range | Log Rng. | Geom. Mean | Titer Conversion per µg/ml | Ref. |
|---|---|---|---|---|---|---|---|---|
| Acute Plasma | IgG ELISA RBD | | 44 | < 100–143,000 | 3.16 | 18,500 | 97.4 | [19] |
| | Neutralizing Act. (FRNT$_{50}$) Live Virus | | 44 | < 50–5,763 | 2.06 | 372 | 1.96 | |
| Acute/Conv. Sera | SPRi IgG RBD Resonance Units | | 32 | 50–4,000 | 1.90 | 2,412 | 12.7 | [20] |
| Acute/Conv. Sera | IgG ELISA ED$_{50}$ | SP | 647 | 1–10,000 | 4.00 | ND | — | [3] |
| | | RBD | 647 | 40–50,000 | 3.10 | ND | — | |
| Conv. Sera | MCLIA IA IgG NP/SP peptide | | 71 | 1.2–181 | 2.26 | 11.3* | 0.063 | [21] |
| Conv. Sera | IgG ELISA U/ml | SP | 59 | 0.12–4,000 | 4.53 | ND | — | [22] |
| | | RBD | 85 | 0.1–200 | 3.30 | ND | — | |
| | Neutr. Act. (NT$_{50}$) | | 118 | 0.5–10,000 | 4.70 | ND | — | |
| Conv. Plasma | IgG ELISA AUC | SP | 149 | 3–9.5x10$^6$ | 0.50 | 4 x 10$^6$ | 21,052 | [6] |
| | | RBD | 149 | 1–7.5x10$^6$ | 0.88 | 2 x 10$^6$ | 10,526 | |
| | Pseudovirus Neutr. (Outpatient) | | 142 | 5–10,000 | 3.30 | 300 | 1.58 | |
| Conv. Plasma | LSA Titer RBD IgG DF | | 20 | 20.7–1,326 | 1.81 | 121 | 0.64 | † |
| Conv./Post-Vax Plasma | IgG ELISA RBD AUC | | 63 | 4,000–400,000 | 2.00 | 30,815 | 162 | [23] |
| | Neutr. Act. (NT$_{50}$) | | 30 | 60–6,000 | 2.00 | 930 | 4.89 | |
| Post-Vax Sera | Multiplex IA IgG AU | SP | 1,155 | 2,000–520,000 | 2.41 | 30,000 | 170 | [13] |
| | | RBD | 1,155 | 1,000–700,000 | 2.85 | 40,000 | 227 | |
| | Pseudovirus Neutr. | ID$_{50}$ | 828 | 10–2,100 | 2.32 | 150 | 0.85 | |
| | Live Virus Neutr. | NF$_{50}$ | 412 | 13–2,900 | 2.35 | 180 | 1.02 | |
| Post-Vax Plasma | IgG ELISA WHO BAU/ml | SP | 59 | 90–2,000 | 1.35 | 566 | 3.22 | [24] |
| | | RBD | 59 | 300–11,000 | 1.56 | 2,138 | 12.1 | |
| | Pseudovirus Neutr. IU/ml | | 59 | 100–5,000 | 1.70 | 1,151 | 6.54 | |
| Post-Vax Plasma | IgG ELISA RBD AUC | | 42 | 30,000–1,050,000 | 1.54 | 241,958 | 1,375 | [25] |
| | Pseudovirus Neutr. NT$_{50}$ | | 42 | 150–15,000 | 2.00 | 1,982 | 11.3 | |

* Median

† The present study.

All of these reports emphasized the importance to further vaccine development and refinement of knowing the protective antibody levels in the peripheral circulation. It will also be important to monitoring the decay of protective immunity after vaccination and relating it to the occurrence of breakout infections. It should be emphasized that measuring antibody binding affinity adds value to assessing the robustness of the antibody response, leading to the conclusion in this study that naturally acquired immunity after SARS-CoV-2 infection is comparable to vaccine-induced immunity. It remains to be determined how the former compares to the latter in terms of persistence. These measures will also enable a realistic assessment of immune evasion by viral variants.

In this study, we utilized previously established quantitative ELISA methods for determination of protective antibody levels relevant to human infectious disease to estimate such levels for SARS-CoV-2 infection (COVID-19), which is characterized by a well-defined virulence factor (the spike protein) and the immune mechanism opposing it (neutralizing anti-SP IgG antibodies). The assays were validated for accuracy, precision and reproducibility by standard accepted criteria. The importance of expressing antibody levels as concentrations of mass per unit volume cannot be overestimated, exemplified by the difficulties of relating one study to another in Table 2. The greatest consistency and correspondence with antibody levels measured in this study were found for virus neutralizing titers.

Expressing antibody levels in terms of concentration also enables relatively straightforward approaches to calculation of protective antibody levels from knowledge of vaccine efficacy. This was especially fortuitous in the case of the Moderna and Pfizer SARS-CoV-2 vaccines, which both exhibited 95% efficacy in preventing infection with the wildtype virus strain. Thus, the lower 95% confidence level of a two-tailed normal geometric distribution of anti-SP IgG antibody concentrations should yield the log of protective concentration.

Although this may appear to be a somewhat simplistic approach, the calculated protective concentration was remarkably similar to conversions to antibody concentrations of analogous 50% protective values calculated from titers of neutralizing antibody [14] and anti-RBD IgG [13] using more complex algorithms and, in fact, fell between those values. Expression of antibody levels as absolute concentrations also permitted calculation of binding affinities by methods previously established for ELISA data [12]. An example of useful information obtained by knowing binding affinities is an indication of preferential high-affinity antibody specificity directed toward the RBD by vaccination, compared to natural infection, which elicits antibodies of equally high affinity specific for non-RBD epitopes (Table 1).

Knowing antibody concentrations and target affinities permits a practical understanding of protective immune mechanisms. For instance, based on a SARS-CoV-2 diameter of 100 nm and spike protein surface density of 25 trimers per virion [17], a concentration of 7.5 µg anti-SP IgG/ml would provide 12 molecules of antibody per spike protein molecule in a viral saturation of a mucosal surface 10 µm in depth [18]. With a binding affinity ($K_D$) of 1.0 nM, 95 percent of SP molecules will be bound by antibody. With the average binding affinity of 0.48 nM found for anti-RBD IgG in fully vaccinated individuals (Table 1), 99% of SP molecules would be bound by antibody. This calculation provides a conceptual framework that is consistent with the empirical data reported here and the foundation for development of a true protective index based on examination of plasma.

The Omicron B.1.1.529 variant first appeared in southern Africa in November 2021 and featured 15–16 amino acid mutations in the RBD alone relative to the prototype SARS-CoV-2 virus [26]. Since then, Omicron subvariants have become the predominant SARS-CoV-2 strains throughout the world. The BA.2.86 subvariant, which appeared in 2023, featured 34 spike protein mutations relative to the BA.2 subvariant [27]. The Omicron subvariants continue to infect cells via the ACE2 receptor [28] and the spike protein affinities for ACE2 are comparable to or even higher than previous SARS-CoV-2 variants [27–30].

Although viral neutralization titers of plasma and sera from vaccinated individuals toward these subvariants were reported to be 1.8- to 22-fold lower than toward the wild type strain [28, 30, 31], infections caused by Omicron subvariants have generally been less severe than those caused by previous variants [32–34]. This observation may be explained by high vaccination rates among the general population, despite lower vaccine efficacy against Omicron subvariants, but it may also indicate that extensive protein mutations in these subvariants are beginning to compromise viral virulence. Correlation of prion-like domain frequency in the spike protein of SARS-CoV-2 variants with virulence [33] tends to support the latter hypothesis, which implies that uncoupling of protective immunity and pathogen virulence is intrinsically limited.

It is of great interest to investigate the protective level of revised vaccine boosters against the various Omicron substrains. The approach explored in this study would be valuable in accomplishing this objective as well as responding to the re-emergence of more virulent strains. The methods utilized in this study would be particularly valuable for revisiting studies of the cellular components of the protective immune response, namely T-dependent B-cell antibody production. Utilizing the recombinant SARS-CoV-2 spike protein as antigen, leukocytes from individuals post-infection and post-vaccination could be interrogated to study the

nature of antibody production, including the evolution of antibody affinities, following challenge. Anti-SP IgG production *in vitro* could then be extrapolated to *in vivo* conditions and compared with measured levels, providing more practical assessments of B-cell memory.

## Conclusions

Quantitative ELISAs have been developed to measure levels of protective IgG antibodies specific for wild type (D614G) SARS-CoV-2 spike protein (SP; $S_1 + S_2$ subunits) in units of absolute concentration (μg/ml) and to determine antibody binding affinities for the SP and its receptor-binding domain (RBD). These assays were used to measure antibody levels and binding affinities in plasma samples from COVID-19 convalescent patients, fully vaccinated subjects and partially vaccinated subjects (n = 21 each). Based on 95% efficacy for the Pfizer and Moderna vaccines, a protective level of 7.5 μg IgG antibody/ml was determined, which agreed well with protective titer levels calculated in two other studies, once converted to concentrations. Protective levels were also determined for two indexes that combined antibody levels with SP and RBD affinities. It is anticipated that these assays will be useful tools for understanding the scope of protective immunity against the SARS-CoV-2 virus.

## Supporting information

**S1 Table. Pearson correlation coefficients between IgG antibody characteristics, n = 42.**
(TIF)

**S2 Table. Comparison of the IgG antibody characteristics between convalescent plasma (CP) and post-vaccination (PV) groups, (n1 = n2 = 21).**
(TIF)

**S3 Table. Mean IgG antibody concentration differences between convalescent plasma (CP) and post-vaccination (PV) groups before and after adjustment for SP affinity ($K_D$) or RBD affinity ($K_D$) or $EC_{50}$ in different GLMs.**
(TIF)

**S1 Fig. Nomograms for conversion of ELISA apparent dissociation constants ($K_D$) to "true" $K_D$s, resulting from Underwood correction calculations as described in Ref. 12.**
(TIF)

**S2 Fig. Superimposability of standard-spiked plasma on a composite standard curve in rSP ELISA.**
(TIF)

**S3 Fig. Polynomial (linear and cubit) correlation of convalescent plasma anti-SP IgG concentration vs LSA anti-RBD titer.**
(TIF)

## Author Contributions

**Conceptualization:** Melvin E. Klegerman, Ira Seferovich, Mohammad H. Rahbar, Manouchehr Hessabi, Amirali Tahanan, Jeffrey D. Cirillo, Dinuka Abeydeera, Steve De Lira.

**Data curation:** Melvin E. Klegerman, Ira Seferovich, Mohammad H. Rahbar, Manouchehr Hessabi, Amirali Tahanan.

**Formal analysis:** Melvin E. Klegerman, Ira Seferovich, Mohammad H. Rahbar, Manouchehr Hessabi, Amirali Tahanan.

**Funding acquisition:** Melvin E. Klegerman, Jeffrey D. Cirillo, David D. McPherson.

**Investigation:** Melvin E. Klegerman, Tao Peng, Ira Seferovich, Mohammad H. Rahbar, Manouchehr Hessabi, Amirali Tahanan.

**Methodology:** Melvin E. Klegerman, Tao Peng, Ira Seferovich, Mohammad H. Rahbar, Manouchehr Hessabi, Amirali Tahanan, Dinuka Abeydeera.

**Project administration:** Melvin E. Klegerman.

**Resources:** Tao Peng, Audrey Wanger, Carolyn Z. Grimes, Luis Z. Ostrosky-Zeichner, Kent Koster, Jeffrey D. Cirillo.

**Software:** Mohammad H. Rahbar, Manouchehr Hessabi, Amirali Tahanan.

**Supervision:** Melvin E. Klegerman, Mohammad H. Rahbar.

**Validation:** Melvin E. Klegerman, Ira Seferovich, Mohammad H. Rahbar, Manouchehr Hessabi, Amirali Tahanan.

**Visualization:** Melvin E. Klegerman, Mohammad H. Rahbar, Manouchehr Hessabi, Amirali Tahanan.

**Writing – original draft:** Melvin E. Klegerman, Ira Seferovich, Mohammad H. Rahbar, Manouchehr Hessabi, Amirali Tahanan.

**Writing – review & editing:** Melvin E. Klegerman, Tao Peng, Ira Seferovich, Mohammad H. Rahbar, Manouchehr Hessabi, Amirali Tahanan, Audrey Wanger, Carolyn Z. Grimes, Luis Z. Ostrosky-Zeichner, Kent Koster, Jeffrey D. Cirillo, Dinuka Abeydeera, Steve De Lira, David D. McPherson.

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
