## [Decision Letter · Decision Letter 0]

1 Jul 2024

PONE-D-24-05130Absolute concentration estimation of COVID-19 convalescent and post-vaccination IgG antibodiesPLOS ONE

Dear Dr. Klegerman,

Thank you for submitting your manuscript to PLOS ONE. After careful consideration, we feel that it has merit but does not fully meet PLOS ONE’s publication criteria as it currently stands. Therefore, we invite you to submit a revised version of the manuscript that addresses the points raised during the review process.

Dear authors,I appreciate the item of your paper and I believe it has a significant scientific value.I recommend you to improve your manuscript following reviewer's suggestions.

We look forward to receiving your revised manuscript.

Kind regards,

Eleonora Nicolai, PhD

Academic Editor

PLOS ONE

“Texas A&M University System, BADAS Study (Bacillus Calmette-Guérin Vaccination as

defense against SARS-CoV-2. A Randomized Placebo-Controlled Trial to Protect Health Care

Workers by Enhanced Trained Immune Responses) (MEK, JDC, KK)

Biostatistics/ Epidemiology/Research Design (BERD) component of the Center for Clinical and

Translational Sciences (CCTS), mainly funded by a grant (UL1 TR003167) from the National

Center for Advancing Translational Sciences (NCATS), awarded to The University of Texas

Health Science Center at Houston (UTHealth) (MHR, MH, AT)”

3. We noted in your submission details that a portion of your manuscript may have been presented or published elsewhere. [An earlier version of this paper was published as a preprint (MedRxiv, 2021; https://doi.org/10.1101/2021.11.19.21266547). PLOS ONE guidelines state that the journal supports publication of preprints.] Please clarify whether this publication was peer-reviewed and formally published. If this work was previously peer-reviewed and published, in the cover letter please provide the reason that this work does not constitute dual publication and should be included in the current manuscript.

Additional Editor Comments:

Dear authors we appreciated the item of your paper, but there are some minor concerns raised by reviewer we would like you to assess.

Reviewers' comments:

Reviewer's Responses to Questions

**Comments to the Author**

1. Is the manuscript technically sound, and do the data support the conclusions?

Reviewer #1: Yes

2. Has the statistical analysis been performed appropriately and rigorously? 

Reviewer #1: Yes

3. Have the authors made all data underlying the findings in their manuscript fully available?

Reviewer #1: No

4. Is the manuscript presented in an intelligible fashion and written in standard English?

Reviewer #1: Yes

5. Review Comments to the Author

Reviewer #1: In the manuscript titled “Absolute concentration estimation of COVID-19 convalescent and post-vaccination IgG antibodies”, the authors have focused on determining absolute antibody concentration and binding affinity in vaccinated and convalescent COVID-19 individuals. The authors have chosen a topic that is very simple at surface level but bridges some important gaps in our knowledge of SARS-CoV-2 immunity. They have determined the antibody concentration and binding affinity with reference to spike protein (SP) and receptor binding domain (RBD) of the spike protein. The study has concluded that the immunity acquired post infection was comparable to vaccine induced immunity. The work done by the authors in this article is significant to the field and the article is competently written. However, there are a few concerns that needs to be addressed:

1. The authors should discuss the findings of the current study in a bit more detail. The discussion portion of the article focuses more on the findings of other research articles and does not discuss the data submitted in this article enough.

2. The authors also need to discuss the future implications of the current study and how it can be useful for any future research endeavors in the field.

6. PLOS authors have the option to publish the peer review history of their article (what does this mean?). If published, this will include your full peer review and any attached files.

Reviewer #1: No

---

## [Author Response · Author response to Decision Letter 0]

5 Sep 2024

We would like to thank the reviewer for helpful comments that have improved the quality of the manuscript. We have amended it in accordance with the comments, as follows.

1. The authors should discuss the findings of the current study in a bit more detail. The discussion portion of the article focuses more on the findings of other research articles and does not discuss the data submitted in this article enough.

We have added three paragraphs to the Discussion discussing the justification for the study design and methodology, evidence for validity of the results, and advantages of the approach.

2. The authors also need to discuss the future implications of the current study and how it can be useful for any future research endeavors in the field.

We have added several sentences at the end of the manuscript describing specific studies that can be performed based on ability to measure actual antibody concentrations.

---

## [Editor Report · Decision Letter 1]

25 Sep 2024

Absolute concentration estimation of COVID-19 convalescent and post-vaccination IgG antibodies

PONE-D-24-05130R1

Dear Dr. Melvin E Klegerman,

We’re pleased to inform you that your manuscript has been judged scientifically suitable for publication and will be formally accepted for publication once it meets all outstanding technical requirements.

Kind regards,

Eleonora Nicolai, PhD

Academic Editor

PLOS ONE
---

## [Editor Report · Acceptance letter]

23 Oct 2024

PONE-D-24-05130R1 

PLOS ONE

Dear Dr. Klegerman, 

I'm pleased to inform you that your manuscript has been deemed suitable for publication in PLOS ONE. Congratulations! Your manuscript is now being handed over to our production team.

Kind regards, 

on behalf of

Dr. Eleonora Nicolai 

Academic Editor

PLOS ONE